# Hyperglycemia Stimulates the Irreversible Catabolism of Branched-Chain Amino Acids and Generation of Ketone Bodies by Cultured Human Astrocytes

**DOI:** 10.3390/biomedicines12081803

**Published:** 2024-08-08

**Authors:** Eduard Gondáš, Eva Baranovičová, Jakub Šofranko, Radovan Murín

**Affiliations:** 1Department of Pharmacology, Jessenius Faculty of Medicine in Martin, Comenius University in Bratislava, Malá Hora 4D, 036 01 Martin, Slovakia; gondas3@uniba.sk; 2Department of Medical Biochemistry, Jessenius Faculty of Medicine in Martin, Comenius University in Bratislava, Malá Hora 4D, 036 01 Martin, Slovakia; sofranko.jakub@gmail.com; 3Biomedical Center Martin, Jessenius Faculty of Medicine in Martin, Comenius University in Bratislava, Malá Hora 4D, 036 01 Martin, Slovakia; eva.baranovicova@uniba.sk

**Keywords:** astrocyte, proton-nuclear magnetic resonance (^1^H-NMR), amino acid, branched-chain 2-oxo acid, branched-chain amino acid, leucine, isoleucine, valine, euglycemia, hyperglycemia, ketone body, metabolomics

## Abstract

Astrocytes are considered to possess a noticeable role in brain metabolism and, as a partners in neuron–glia cooperation, to contribute to the synthesis, bioconversion, and regulation of the flux of substrates for neuronal metabolism. With the aim of investigating to what extent human astrocytes are metabolizing amino acids and by which compounds are they enriching their surroundings, we employed a metabolomics analysis of their culture media by ^1^H-NMR. In addition, we compared the composition of media with either 5 mM or 25 mM glucose. The quantitative analysis of culture media by ^1^H-NMR revealed that astrocytes readily dispose from their milieu glutamine, branched-chain amino acids, and pyruvate with significantly high rates, while they enrich the culture media with lactate, branched-chain keto acids, citrate, acetate, ketone bodies, and alanine. Hyperglycemia suppressed the capacity of astrocytes to release branched-chain 2-oxo acids, while stimulating the generation of ketone bodies. Our results highlight the active involvement of astrocytes in the metabolism of several amino acids and the regulation of key metabolic intermediates. The observed metabolic activities of astrocytes provide valuable insights into their roles in supporting neuronal function, brain metabolism, and intercellular metabolic interactions within the brain. Understanding the complex metabolic interactions between astrocytes and neurons is essential for elucidating brain homeostasis and the pathophysiology of neurological disorders. The observed metabolic activities of astrocytes provide hints about their putative metabolic roles in brain metabolism.

## 1. Introduction

Astrocytes are central to brain metabolism, performing essential tasks including energy production, neurotransmitter regulation, detoxification, and maintaining homeostasis [1,2,3,4,5]. These diverse roles ensure that the brain’s metabolic demands are met, neuronal function is supported, and the brain environment remains stable and protected from damage. The brain relies on a limited number of substances imported from the bloodstream to support its metabolism. The specificity and capacity of transporters expressed at the blood–brain barrier restrict the availability of polar and ionic substrates for brain metabolism [6]. Among these substrates, glucose is the primary energy source, with several glucose transporters maintaining its import capacity, surpassing that of all other compounds [7].

Recent data suggest that, in addition to glucose, amino acids can also serve as fuel molecules in energy production, playing a significant role in sustaining brain metabolism under various pathological and experimental conditions. Among these amino acids, neutral amino acids have a high importing capacity through specific transporters [8,9,10], implicating their involvement in brain metabolism [11,12,13,14,15,16]. Specifically, leucine, isoleucine, and valine are the most effectively imported into the human brain parenchyma. These three essential amino acids, known as branched-chain amino acids (BCAAs), have a common structural motif and serve several critical functions in the brain. BCAAs act as monomers for protein synthesis and nitrogen donors for the de novo synthesis of glutamate and GABA, and they balance the glutamate/glutamine cycle. Additionally, BCAAs can serve as alternative sources of acetyl-CoA and propionyl-CoA, contributing to brain anaplerotic, anabolic, and oxidative metabolism [11,12,13,14,15,16,17,18].

The biochemical aspects of astrocytic metabolism have been studied using animal models, animal cell cultures, and human cancer cell lines. Studies with cultured rat astrocytes have shown that these cells can catabolically convert BCAAs into several molecules [19,20,21], including ketone bodies [22]. The metabolic capacity of rat astrocytes correlates well with the expression of enzymes specific to BCAA catabolic pathways, observed in both cultured astrocytes and in situ [23,24,25].

Hyperglycemia, a major comorbidity of untreated diabetes, can affect brain metabolism [26,27,28], including the metabolism of amino acids [29,30]. It is hypothesized that hyperglycemia-induced changes in amino acid levels in the brain parenchyma may result from altered astrocytic metabolism. To investigate the extent to which increased glucose levels affect human astrocyte metabolism, we employed an ^1^H-NMR analysis. This technique allowed us to identify changes in the ability of cultured human astrocytes to utilize components from their culture media and subsequently enrich their milieu with metabolic products. Our results revealed that human astrocytes, regardless of glucose levels, remove glutamine, BCAAs, and pyruvate from their medium. Additionally, they enrich the culture medium with several metabolic intermediates, including lactate, citrate, acetate, alanine, branched-chain keto acids (BCKAs), and ketone bodies. Experimental hyperglycemia influenced the ability of cultured human astrocytes to release BCKAs and ketone bodies into the culture medium. Estimating the ratio of BCKA to BCAA, which reflects the rate for BCKA entry into the irreversible part of BCAA catabolism, revealed that hyperglycemia supports the catabolism of carbon skeletons originating from BCAAs. Given that BCAAs and their cognate BCKAs play roles in both the synthesis of glutamate and the glutamate/glutamine cycle, the hyperglycemic effect on BCAA catabolism may contribute to the molecular changes underlying the pathoneurological aspects of hyperglycemia.

## 2. Materials and Methods

### 2.1. Chemicals

Nicotinamide adenine dinucleotide (NAD^+^), and 3-hydroxybutyrate dehydrogenase from Rhodobacter sphaeroides were purchased from Roche (Basel, Switzerland). Ethylenediaminetetraacetic acid (EDTA) was purchased from Serva (Fischer Scientific, Göterberg, Sweden). The DC Protein assay kit was from Bio-Rad Laboratories, Hercules, CA, USA. All other chemicals—namely, glycine, hydrazine, lactate, 3-hydroxybutyrate, sodium 3-(trimethylsilyl)propionic-2,2,3,3-D_4_, Dulbecco’s phosphate-buffered saline (DPBS), Triton X-100, and bovine serum albumin—were purchased from Sigma (Sigma St. Louis, MO, USA).

### 2.2. Cell Culture

In this study, we used normal human astrocytes (NHAs) obtained from the Lonza Group (Basel, Switzerland, Cat. No. CC-2565). The human astrocyte cells were cultured in Dulbecco’s Modified Eagle Medium (Gibco, Waltham, MT, USA; 10569010), with a glucose concentration of 25 mM supplemented with 10% (*v*/*v*) One Shot Fetal Bovine Serum (Gibco, Waltham, MT, USA), 1% (*v*/*v*) N-2 Supplement, 100 U/mL penicillin, and 0.1 mg/mL streptomycin sulfate, which is a specific medium for astrocytes growth, at 37 °C, in an incubator containing 5% CO_2_. The culture medium was renewed every two days. The trypsinizations were used for passaging after the cells had reached an approximate confluency of 80–90%, using 0.5% trypsin solution.

### 2.3. ^1^H-NMR Analysis

Astrocyte cells, in the 5th passage, were cultured in 6-well plates to confluency. For the experiment, the 1 mL of the culture medium per well was replaced with Dulbecco’s Modified Eagle Medium (DMEM) with a 25 mM (Sigma, D6429) or 5 mM (Sigma, D6046) glucose level, which was enriched with 10% fetal bovine serum and cultured at 37 °C for 24 h. After incubation for 24 h, the medium was collected, centrifuged at 10,000× *g* and stored at −80 °C until analysis.

For the ^1^H-NMR analysis, 500 µL of culture medium was mixed with 100 µL of NMR solution (consisting of 200 mM phosphate buffer with pH 7.4 in D_2_O, enriched with [sodium 3-(trimethylsilyl)propionic-2,2,3,3-D_4_] (TMS-D_4_) to a level of 0.2 mM) and transferred to an NMR tube. The ^1^H-NMR spectra were recorded on a Bruker Avance III, 600 MHz, equipped with a TCI cryoprobe, and subsequently processed as already described [31]. The concentration of metabolites in the culture medium before incubation was determined using external standards, where the measured concentration was calculated through a linear relation to the signals of known concentration.

#### Cell Lysate Preparation

After 24 h incubation, the culture medium was discharged, and the attached cells were immediately washed three times with 4 °C DPBS. Subsequently, the cells were lysed with 4 °C lysis buffer consisting of 50 mM TRIS/HCl with pH 7.5, 1 mM EDTA, and 1% (*w*/*V*) Triton X-100. A rubber cell scraper was used to scrape the attached cells. The lysates were transferred into sterilized tubes and centrifuged at 10,000 rpm for 10 min

### 2.4. Protein Estimation

For measurement of the total protein concentration in the generated lysates, the commercially available DC Protein assay kit (Bio-Rad Laboratories, Hercules, CA, USA) was used. Bovine serum albumin was used as a standard. The measurement of absorbance was performed according to the manufacturer’s instructions using a BioTek Synergy H4 hybrid microplate reader (Bio Tek, Winooski, VT, USA).

### 2.5. Cell Survival

The lactate dehydrogenase (LDH) enzymatic assay was performed to calculate the total and specific activity of LDH.

Reaction buffer 1 (RB1) consisted of a mixture of 0.1 M glycine/NaOH buffer with pH 9, with the addition of 0.05 M hydrazine, 5 mM NAD^+^, and 10 mM lactate. For the assay, 30 μL of centrifuged medium or centrifuged lysate was mixed with 270 μL of RB1 in the 96-well plate. NADH formation in time was recorded at the absorbance (λ = 340 nm). The measured data were used to calculate the enzymatic activity of LDH in the medium or lysates.

The cell viability was calculated as the ratio of LDH activity in the lysates to the total LDH activity according to method previously described [32]. The total LDH activity is the sum of LDH activities estimated in lysates and media.

### 2.6. Enzymatic Estimation of 3-Hydroxybutyrate Level

The measurement of the amount of released 3-hydroxybutyrate (3-OHB) in the culture medium was performed as already described [31]. Briefly, the reaction buffer (RB2) consisted of a mixture of 0.1 M glycine/NaOH buffer with pH 9, with the supplementation of 0.05 M hydrazine, 5 mM NAD^+^, and 3 µg/mL of purified 3-hydroxybutyrate dehydrogenase from Rhodobacter sphaeroides. A total of 30 μL of clarified supernatant prepared from denaturated and centrifuged culture medium was mixed with 270 μL of RB2, and the solution was immediately used for the measurement of the generated NADH, which was monitored at 340 nm by a Synergy H4 microplate reader (Bio Tek, Winooski, VT, USA). The absorbance was monitored until the measured values reached a plateau lasting at least 5 min. The calibration curve from increasing concentrations of the 3-OHB solution in the phosphate buffer was used to quantify the 3-OHB released to the culture medium. Subsequently, the capability of astrocyte cells to release 3-OHB into the culture medium was determined by considering the molar amount of released 3-OHB per mg of cellular proteins over a precisely defined incubation duration (24 h).

### 2.7. Statistical Analysis

The results represent the mean ± SD or ±SEM of six independent experiments, which consist of six biological replicates of the cells in the same passage number. A one-way analysis of variance with post hoc comparisons using the Student–Newman–Keuls test was performed to test for differences between the test groups. A statistical analysis was performed using InStat software v. 3.10 (GraphPad Software, Boston, MA, USA).

## 3. Results

To demonstrate the effect of higher (25 mM) and normal (5 mM) glucose concentrations in culture medium on cultured human NHA astrocytes cells, the culture media were subjected to metabolic analysis after a 24 h incubation. We found that a concentration of 5 or 25 mM glucose in the culture medium had no significant effect on the total protein content of lysates from NHA cells, cell survival, and the specific activity of lactate dehydrogenase in lysates (Table 1). Cell survival was calculated as the ratio of activity the lactate dehydrogenase in the culture medium released from dead cells to the sum of the activities of lactate dehydrogenase in the medium and lysates.

To assess the capability of astrocyte cells to metabolize amino acids and other metabolites from their culture medium, the culture medium with 25 mM or 5 mM glucose concentration was collected after a 24 h incubation and subjected to ^1^H-NMR analysis. The spectra obtained from the ^1^H-NMR chromatogram (Figure 1) were used to calculate the residual concentrations of the metabolites composing the medium and the metabolites released into the culture medium (Table 2). The concentration of metabolites was determined by measuring their levels in the culture medium before incubation with cells against external standards. To estimate the specific uptake and release of some metabolites from or into the culture medium, the data obtained from the ^1^H-NMR analysis were standardized to total protein content and incubation time (nmol·mg^−1^·h^−1^; Figure 2). Our data show that branched-chain amino acids (BCAA) are mostly removed from the culture medium compared to other amino acids (Table 2), and significant changes in the uptake of BCAA from the culture media caused by higher glucose levels were not observed (Figure 2A). Simultaneously, the presence of peaks specific for branched-chain keto acids (BCKAs, Figure 1B,C) among the obtained spectra points out the capability of cultured astrocyte cells to release BCKA into the culture medium (Figure 2B). The incubation of cells with 25 mM glucose revealed a significant decrease in the release of two BCKAs, namely, α-ketoisocaproate (KIC), and α-keto-methylvalerate (KMV), into the culture medium compared to 5 mM glucose. In addition, the appearance of specific peak for acetone (Figure 1D), representing ketone body, was identified in the spectra, with a higher intensity in the culture medium with 25 mM glucose (Figure 2C). The calculated ratios of released KIC to taken up leucine, KMV to isoleucine, and BCKA to BCAAs indicate that 50% ± 10% of leucine molecules and 46% ± 7% of isoleucine molecules taken up from the culture medium were released as KIC or KMV, respectively, in the case of 5 mM glucose concentration in the culture medium, while 25 mM glucose level in culture medium decreased both ratios KIC/Leu and KMV/Ile to 35% ± 4% or 33% ± 7%, respectively (Figure 2D). Indeed, the higher glucose concentration (25 mM) significantly also decreased the ratio of BCKA to BCAA, 48% ± 8%, in comparison to a normal (5 mM) glucose concentration in the culture medium, with a ratio of 64% ± 11% (Figure 2D). Therefore, it could be assumed that in the case of higher glucose levels, a large amount of BCAA molecules undergo further intracellular metabolism.

The obtained ^1^H-NMR spectra did not allow the identification of the signal of two remaining ketone bodies, 3-hydroxybutyrate (3-OHB) and acetoacetate, and also their quantification. For this purpose, the spectrophotometric method for the quantification of 3-hydroxybutyrate was performed. The estimation of the specific release of 3-OHB was calculated in the same way as the ^1^H-NMR method; thus, the data obtained from the spectrophotometric analysis were standardized for the total protein content and incubation time (nmol*mg^−1^*h^−1^; Table 2). This analysis showed that the specific release of 3-OHB into the culture medium was significantly increased during 24 h incubation with 25 mM glucose, in comparison to 5 mM glucose, in the culture medium (Figure 2C).

## 4. Discussion

The ^1^H-NMR analysis on the media from the primary culture of human astroglial cells revealed the capability of the cells to actively modify the composition of the media by concurrent elimination and enrichment with several compounds. In addition, our study demonstrates that hyperglycemia significantly influences the metabolism of BCKAs and ketone bodies by astrocytes. These findings are relevant for deducing the metabolic role of astrocytes in the human brain and, furthermore, the metabolic responses of the astroglial cells to hyperglycemia.

Under experimental euglycemic conditions, astrocytes efficiently removed glucose, BCAAs, glutamine, and pyruvate from the medium, while enriching it with intermediates of cellular metabolism, such as lactate, citrate, acetate, alanine, BCKAs, and ketone bodies. This metabolic profile underscores the versatility of astrocytes in utilizing multiple substrates to sustain their energy and biosynthetic demands, thereby contributing to the overall metabolic homeostasis of the brain in euglycemic conditions.

Astrocytes are recognized for their crucial role in brain energy metabolism, particularly in the conversion of glucose to lactate [33,34,35,36,37,38]. Studies with animal models have demonstrated that rodent astrocytes possess a high specific capacity of lactate release [39], surpassing the estimated values for cultured human astroglial cells (Table 2). Nonetheless, the observed capability of human astroglial cells to release lactate supports the hypothesis of an astrocyte–neuron lactate shuttle, which supports the energy metabolism of neurons [40]. Additionally, the astrocytic release of lactate may serve signaling roles in the brain [41].

Although human astrocytes can import pyruvate from the extracellular space (Table 2), the estimated quantity of its import accounts for only a fraction of the total amount of released lactate. Similar to animal astrocytes, human astroglial cells are expected to generate most of the lactate through the anaerobic glycolysis of glucose [38]. Indeed, the cultured human astroglial cells readily disposed of glutamine, an amino acid with several distinct functions in cellular metabolism and the synthesis of signaling molecules. Glutamine’s structure allows it to contribute to the cellular metabolism of nitrogen-containing compounds. Additionally, brain glutamine is essential for sustaining the levels of glutamate and gamma-aminobutyrate in the brain parenchyma [42]. Furthermore, the carbon skeleton of glutamine can play an anaplerotic role and may enter energetic metabolism, being oxidized to carbon dioxide in mitochondria [33].

The catabolism of glutamine can provide the amino group for the transamination of pyruvate to alanine. The latter may be released from astrocytes, to remove the superfluous nitrogen from astrocytes, to exchange the amino group with neurons, or to support neuronal mitochondrial metabolism [43,44]. The analyzed media were also enriched with pyroglutamate, a compound that can be generated either spontaneously through the cyclization of glutamate and glutamine or enzymatically from glutathione [45].

Moreover, human astroglial cells dispose of a substantial amount of BCAAs, and their irreversible catabolism appeared to be glucose-dependent. Hyperglycemia enhances the disappearance of BCKAs and increases the release of ketone bodies into the culture medium. This suggests that human astrocytes may be capable of endogenously generating ketone bodies in the brain parenchyma. Even though the previous data showed that cultured rat astroglial cells generate substantial amounts of ketone bodies from leucine [22] under hyperglycemic conditions, the ability of astrocytes to generate ketone bodies under euglycemic conditions remained an open question.

The elevated production and release of ketone bodies under hyperglycemia highlight the metabolic plasticity of human astrocytes. Ketone bodies, typically produced during states of a decreased glucose level or insulin availability, are now synthesized in response to hyperglycemia, indicating the robust metabolic adaptability of astrocytes to varying glucose levels. This ketogenesis may provide an auxiliary energy source for neighboring neurons [46], supporting neuronal function and contributing to intercellular communication between glia and neurons [46,47].

In contrast to BCAAs, the levels of three essential amino acids, namely, lysine, methionine and tyrosine, were significantly increased in media with euglycemic conditions. The levels of the remaining essential amino acids did not change significantly either in euglycemic or hyperglycemic media (Table 2). The possibility of increased levels of some of the essential amino acids in culture media from astrocytes has already been documented [22], but not studied in detail. However, it could be hypothesized that, similar to human glioma cells, astrocytes are also capable of using extracellular proteins as the putative source of amino acids [48]. Another aspect that may affect the overall metabolism of essential amino acids could be the proteosynthetic ratio of the cells. Since we used confluent cultures of astrocytes, the growth and dividing of the cells may be expected to be restricted by the contact inhibition of proliferation [49], and therefore, the uptake of those essential amino acids that are not further catabolized may be minimal.

In conclusion, our study elucidates the dynamic role of astrocytes in maintaining brain metabolic homeostasis through substrate flexibility and adaptive metabolic responses. Our results underscore the importance of astrocytes not only as supportive glial cells but as active participants in brain metabolism, capable of responding to and modulating metabolic challenges. These findings enhance our understanding of astrocytic contributions to brain metabolism, particularly under pathological conditions such as hyperglycemia. Future research should aim to delineate the signaling pathways and regulatory mechanisms underlying these metabolic adaptations, which could uncover potential therapeutic targets for metabolic disorders affecting the brain.

## Figures and Tables

**Figure 1 biomedicines-12-01803-f001:**
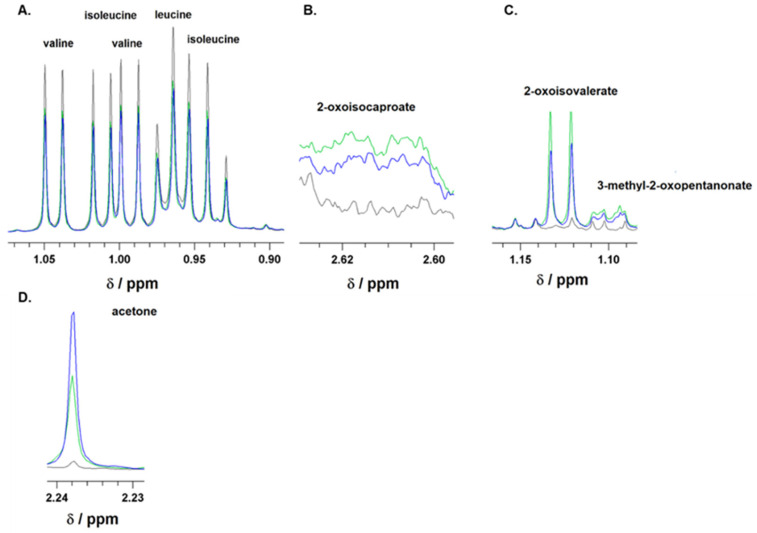
Representative ^1^H-NMR spectra of branched-chain amino acids (BCAAs), leucine, isoleucine, and valine (**A**); branched-chain keto acids (BCKAs), 2-oxoisocaproate (**B**), 2-oxoisovalerate, and 3-methyl-2-oxopentanonate (**C**), as well as of acetone (**D**). Spectra were recorded for culture media supplemented with 10% FBS before incubation (black line) or after 24 h incubation of astrocytes in media either with 5 mM glucose (green line) or 25 mM (blue line) initial glucose level. The ranges of chemical shifts (δ) for BCAA, BCKA, and acetone are depicted.

**Figure 2 biomedicines-12-01803-f002:**
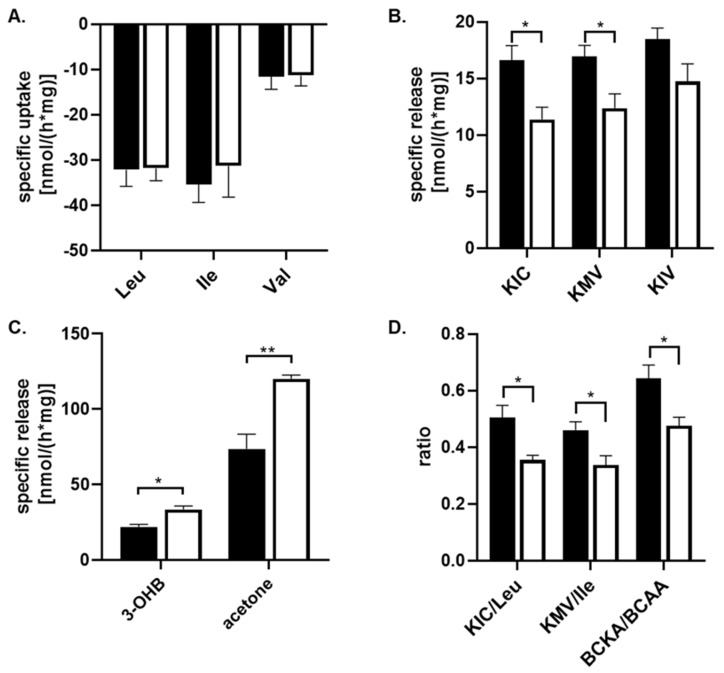
The quantification of the specific uptake or release of metabolites from culture medium either with 5 mM glucose (black column) or with 25 mM (white column) supplemented with 10% FBS on cultured human astrocytes by ^1^H-NMR analysis, except 3-hydroxybutyrate (3-OHB), which has been estimated enzymatically. Both groups were incubated for 24 h. The specific release (negative value) and uptake of the metabolites from the culture medium were quantified by standardizing them to the total cellular protein mass and incubation time (nmol*h^−1^*mg^−1^). The estimation of the specific uptake of leucin (Leu), isoleucine (Ile), and valine (Val) (**A**) from the culture medium; the specific release of BCKA, namely, α-ketoisocaproate (KIC), α-keto-methylvalerate (KMV), and α-ketoisovalerate (KIV) (**B**); the specific release of 3-OHB, and acetone (**C**) into the culture medium; and the calculation of KIC/leu, KMV/Ile, and BCKA/BCAA ratios (**D**) are depicted. All values represent the mean ± SEM from six independent experiments, and * represents the value of *p* ≤ 0.05, and ** *p* ≤ 0.01.

**Table 1 biomedicines-12-01803-t001:** Effect of normoglycemia (5 mM) and hyperglycemia (25 mM) on total lysate protein per well, cell survival, and specific activity of lactate dehydrogenase (a_s_ LDH) on human astrocytes. Both groups were incubated for 24 h in culture medium. All values represent mean ± SD from six independent experiments.

Parameter (Unit)	Glucose	
	5 mM	25 mM
Protein content (mg)	0.11 ± 0.03	0.09 ± 0.03
Cell survival (%)	70 ± 5	67 ± 6
a_s_ [LDH] (U/g)	266 ± 29	231 ± 50

**Table 2 biomedicines-12-01803-t002:** The level of metabolites in culture DMEM/FCS prior or after 24 h incubation in the presence of cultured human astrocytes. To simulate euglycemia or hyperglycemia, the medium at the beginning of incubation contained glucose at a level of either 5 mM or 25 mM, respectively. The concentrations of compounds were estimated either by ^1^H-NMR analysis, or enzymatically. All values represent mean ±SD from six independent experiments. The statistical analysis compared a 5 mM glucose concentration in the culture medium against 25 mM, and * represents the value of *p* ≤ 0.05, and ** *p* ≤ 0.01, *** *p* ≤ 0.001.

Concentration (µM)	Medium	Glucose	
		5 mM	25 mM
lactate	801 ± 38	2195 ± 176 **	1862 ± 60
pyruvate	917 ± 26	341 ± 27 **	430 ± 27
citrate	18.5 ± 0.5	25 ± 1	26 ± 1
acetate	90 ± 12	136 ± 14	128 ± 40
acetone	29 ± 2	174 ± 78	253 ± 98
3-hydroxybutyrate	0	56 ± 2	62 ± 5
glutamine	5045 ± 180	3878 ± 136 *	3716 ± 50
leucine	793 ± 15	714 ± 18	727 ± 7
α-ketoisocaproate	22 ± 4	61 ± 5 **	45 ± 3
isoleucine	751 ± 19	663 ± 24	674 ± 12
α-keto-methylvalerate	14 ± 2	53 ± 4 ***	39 ± 2
valine	773 ± 10	745 ± 14	748 ± 16
α-ketoisovalerate	7 ± 1	48 ± 4 **	35 ± 2
alanine	148 ± 2	206 ± 6 **	190 ± 3
histidine	464 ± 8	467 ± 12	465 ± 11
lysine	867 ± 40	1024 ± 21 *	980 ± 68
methionine	209 ± 3	222 ± 5 **	210 ± 2
phenylalanine	470 ± 6	487 ± 8	482 ± 8
threonine	1016 ± 34	1043 ± 41	1014 ± 41
tryptophan	21 ± 1	23 ± 2	23 ± 1
tyrosine	367 ± 6	381 ± 7 *	371 ± 6

## Data Availability

The original contributions presented in the study are included in the article, further inquiries can be directed to the corresponding author.

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
