# Peer review of "Hyperglycemia Stimulates the Irreversible Catabolism of Branched-Chain Amino Acids and Generation of Ketone Bodies by Cultured Human Astrocytes"

_biomedicines, 2024, doi:10.3390/biomedicines12081803_

Round 1
Reviewer 1 Report
Comments and Suggestions for Authors
The authors have demonstrated that hyperglycemia inhibits the ability of astrocytes to release branched-chain amino acids while stimulating the production of ketone bodies. Their findings highlight that astrocytes are actively involved in the metabolism of several amino acids, as well as the regulation of key metabolic intermediates. However, this study only observed the altered metabolism response to hyperglycemia. What are the mechanisms underlying the altered metabolism? What is the effect of altered metabolism on neurons have not been clarified. Additional experiments are suggested to clarify underlying mechanisms and their effect on neurons.
Comments on the Quality of English LanguageMinor editing of English language required。
Author Response
comment 1:
The authors have demonstrated that hyperglycemia inhibits the ability of astrocytes to release branched-chain amino acids while stimulating the production of ketone bodies. Their findings highlight that astrocytes are actively involved in the metabolism of several amino acids, as well as the regulation of key metabolic intermediates. However, this study only observed the altered metabolism response to hyperglycemia. What are the mechanisms underlying the altered metabolism? What is the effect of altered metabolism on neurons have not been clarified. Additional experiments are suggested to clarify underlying mechanisms and their effect on neurons.
Responde 1:
We are very grateful for the reviewer ´s comments and proposed suggestions. As raised by reviewer we have showed that several alternative substrates can enter the metabolism of human cultured astrocytes and the cellular metabolism can be subject of alteration in presence of high glucose level. We fully agree with reviewer´s comment, that this knowledge together with the possibility to exploit such cellular model of in vitro stimulated hyperglycemia could be fundamental for further studies dealing with the identification of the molecular mechanism underlying observed phenomenon.
Reviewer 2 Report
Comments and Suggestions for Authors
The authors of the manuscript "Hyperglycemia Stimulates the Irreversible Catabolism of 2 Branched-Chain Amino Acids and Generation of Ketone Bodies by Cultured Human Astrocytes" have researched the features of the influence of hyperglycemia on the astrocyte release/uptake of compounds and metabolites from the culture medium. Since the number of patients with diabetes mellitus is currently steadily increasing, it is of undeniable interest to study the effect of hyperglycemia in uncompensated diabetes on the function of astrocytes. In this view, the presented manuscript has certain scientific interest.
Comments.
1. In the chapter “2. Materials and Methods” in 2.2. It is necessary to indicate the glucose concentration in the medium in which the cells are cultured.
2. It is not clear at what passage the cells were taken into the experiment, how many days were they cultured on the plate before the start of the experiment? With what concentration of cells and in what volume were they placed per well of a 6-well plate?
3. In "2.4. Protein Estimation" needs to clarify: What was the lysbuffer was used to receive cell lysate?
4. In "2.7. Statistical Analysis" the authors have written “...of six independent experiments..”, It is necessary to clarify whether these were separate passages or plates, if passages, then which ones? The range or number of passages used in the study must be specified.
5. Lines 159-161: The authors have provided descriptions of the calculation of cell death using the LDH test. This information needs to be moved from chapter "3. Results" to the appropriate methods section.
6. In Table 1 on the page 5, the authors have written “Protein content (mg).” It is needed to clarify the value of measuring - mg per ml or per well?
7. Lines 34, 198: The authors have used the word “euglycemia”, How is it different from “normoglycemia” (Lines 193, 213) It is necessary to use uniform terms and designations.
8. Lines 240-245 276, 286, 288: The authors have used the term “euglycemic”, Please see remarks up (7.).
9. In Fig. 1 authors must correctly name the components of the legend. All three groups are culture medium, not 5mM/25mM glucose, as indicated in the figure legend.
11. It is necessary to make the changes in Fig. 2 to correctly reflect the data. In the caption to the figure and in the legend, it is necessary to clarify what 5 mM and 25 mM are - specifying that this is the glucose content in the culture medium or normo- or hyperglicemia.
12. The medium used in the work for culturing astrocytes contains both a specific amino acid composition and 10% serum, which has a complex composition. In this regard, have the authors analysed what happens to the composition of this medium when it is kept in an incubator for 24 hours in the absence of cells? Accordingly, as a control, it would be more correct to use a medium with a corresponding glucose concentration of 5 or 25 mM, which was kept in an incubator without cells for 24 hours. Since it is not clear how stable the composition of the medium is when kept for a long time at 37 degrees and 5% CO2.
13. In the abstract, the authors have written: “The observed metabolic activities of astrocytes provide valuable insights into their roles in supporting neuronal function, brain metabolism, and intercellular metabolic interactions within the brain.” However, the work did not study two-dimensional cultures, including both astrocytes and neurons, so it is not correct to talk about the state of neurons and intercellular interaction.

Author Response
- In the chapter “2. Materials and Methods” in 2.2. It is necessary to indicate the glucose concentration in the medium in which the cells are cultured.
- We have added this information to our MS:
“The human astrocyte cells were cultured in a Dulbecco´s Modified Eagle Medium (Gibco, 10569010) with glucose concentration 25 mM ....”
- It is not clear at what passage the cells were taken into the experiment, how many days were they cultured on the plate before the start of the experiment? With what concentration of cells and in what volume were they placed per well of a 6-well plate?
- We have added the information about passage in the manuscript (5th).
- The cells were cultured to confluency for this experiment (which last up to 3 days).
- The cells were seeded in recommended density of 4 × 104 cells/cm2 (360,000 in 2–3 mL in one well of a six well plate.
- While cells were grown in 3 ml per well of culture medium. For experiment, the medium was removed and replaced with fresh culture medium with 25- or 5-mM glucose level for 24h. The volume of the medium used for 24 h experimental incubation was 1 ml per well of a 6-well plate.
- In "2.4. Protein Estimation" needs to clarify: What was the lys buffer was used to receive cell lysate?
- We would like to thank you for this comment. We have added the paragraph about preparation of lysates to Material and methods part in our manuscript.
- In "2.7. Statistical Analysis" the authors have written “...of six independent experiments..”, It is necessary to clarify whether these were separate passages or plates, if passages, then which ones? The range or number of passages used in the study must be specified.
- We have added the information about statistical analysis to MS: “The results represent the mean ±SD, or ±SEM of six independent experiments, which consist of six biological replicates of the cells in the same passage number “. Indeed, we have added the information about passage number (5th) in the paragraph on 1H-NMR analysis.
- Lines 159-161: The authors have provided descriptions of the calculation of cell death using the LDH test. This information needs to be moved from chapter "3. Results" to the appropriate methods section.
- We have moved this information from result part to the methods.
- In Table 1 on the page 5, the authors have written “Protein content (mg).” It is needed to clarify the value of measuring - mg per ml or per well?
- We have clarified the protein content in our manuscript.
- Lines 34, 198: The authors have used the word “euglycemia”, How is it different from “normoglycemia” (Lines 193, 213) It is necessary to use uniform terms and designations.
- Normoglycemia and euglycemia are synonyms, so we have uniformed the term to normoglycemia.
- Lines 240-245 276, 286, 288: The authors have used the term “euglycemic”, Please see remarks up (7.).
- Normoglycemia and euglycemia are synonyms, so we have uniformed the term to normoglycemia.
- In Fig. 1 authors must correctly name the components of the legend. All three groups are culture medium, not 5mM/25mM glucose, as indicated in the figure legend.
- We have changed the Fig.1 and clarified the components of the legend.
- It is necessary to make the changes in Fig. 2 to correctly reflect the data. In the caption to the figure and in the legend, it is necessary to clarify what 5 mM and 25 mM are - specifying that this is the glucose content in the culture medium or normo- or hyperglycemia.
- We have changed the Fig.2 and clarified the components of the legend.
- The medium used in the work for culturing astrocytes contains both a specific amino acid composition and 10% serum, which has a complex composition. In this regard, have the authors analysed what happens to the composition of this medium when it is kept in an incubator for 24 hours in the absence of cells? Accordingly, as a control, it would be more correct to use a medium with a corresponding glucose concentration of 5 or 25 mM, which was kept in an incubator without cells for 24 hours. Since it is not clear how stable the composition of the medium is when kept for a long time at 37 degrees and 5% CO2.
- We are really grateful for this comment. We have to agree that this is a perfect suggestion for additional control. We will include it as a control in designing our future experiments. Unfortunately, at this moment we have to admit, that our recent financial and technical conditions do not allow us to complement the recent data set with the results of suggested analysis.
- In the abstract, the authors have written: “The observed metabolic activities of astrocytes provide valuable insights into their roles in supporting neuronal function, brain metabolism, and intercellular metabolic interactions within the brain.” However, the work did not study two-dimensional cultures, including both astrocytes and neurons, so it is not correct to talk about the state of neurons and intercellular interaction.
- Thank you for pointing down our attention to this opinion. We have amended the final part of the abstract in accordance with the suggestion.
Round 2
Reviewer 1 Report
Comments and Suggestions for Authors
Additional experiments are suggested to clarify underlying mechanisms of the altered metabolism and their effect on neurons.
Comments on the Quality of English LanguageNo.
Author Response
comment 1: Additional experiments are suggested to clarify underlying mechanisms of the altered metabolism and their effect on neurons.
feedback 1: We are really grateful for this comment. We have to agree that this is a perfect suggestion for future research. Unfortunately, at this moment we have to admit, that our recent financial and technical conditions do not allow us to implement mentioned experiments.
Reviewer 2 Report
Comments and Suggestions for Authors
The manuscript has been improved and changes have been made. The authors' responses to comments are correct. In the presented form, the manuscript can be accepted for publication.

Author Response
The manuscript has been improved and changes have been made. The authors' responses to comments are correct. In the presented form, the manuscript can be accepted for publication.
We would like to thank you for your possitive and excelent review.